# Novel Strategy for Alzheimer's Disease Treatment through Oral Vaccine Therapy with Amyloid Beta

**Yasunari Matsuzaka [1,2,\*] and Ryu Yashiro [2,3]**

1 Division of Molecular and Medical Genetics, Center for Gene and Cell Therapy, The Institute of Medical Science, The University of Tokyo, Minato-ku, Tokyo 108-8639, Japan

2 Administrative Section of Radiation Protection, National Institute of Neuroscience, National Center of Neurology and Psychiatry, Kodaira, Tokyo 187-8551, Japan

3 Department of Infectious Diseases, Kyorin University School of Medicine, 6-20-2 Shinkawa, Mitaka-shi, Tokyo 181-8611, Japan

\* Correspondence: yasunari80808@ims.u-tokyo.ac.jp; Tel.: +81-3-5449-5372

**Abstract:** Alzheimer's disease (AD) is a neuropathology characterized by progressive cognitive impairment and dementia. The disease is attributed to senile plaques, which are aggregates of amyloid beta (Aβ) outside nerve cells; neurofibrillary tangles, which are filamentous accumulations of phosphorylated tau in nerve cells; and loss of neurons in the brain tissue. Immunization of an AD mouse model with Aβ-eliminated pre-existing senile plaque amyloids and prevented new accumulation. Furthermore, its effect showed that cognitive function can be improved by passive immunity without side effects, such as lymphocyte infiltration in AD model mice treated with vaccine therapy, indicating the possibility of vaccine therapy for AD. Further, considering the possibility of side effects due to direct administration of Aβ, the practical use of the safe oral vaccine, which expressed Aβ in plants, is expected. Indeed, administration of this oral vaccine to Alzheimer's model mice reduced Aβ accumulation in the brain. Moreover, almost no expression of inflammatory IgG was observed. Therefore, vaccination prior to Aβ accumulation or at an early stage of accumulation may prevent Aβ from causing AD.

**Keywords:** Alzheimer's disease; amyloid β protein; amyloid precursor protein; oral immune tolerance; vaccine therapy

## 1. Introduction

Dementia is a general term for the loss of memory and other intellectual abilities, causing hindrance in daily life activities [1,2]. Alzheimer's dementia is the most common type of dementia, which accounts for 60–80% of all the cases of dementia and occurs due to the progressive degeneration of cranial nerves and partial atrophy of the brain [3,4]. Alzheimer's disease (AD), from its early stage of onset, is insidious, and begins damaging the hippocampus, which controls memory in the brain; the first manifestation of AD is a decline in judgment [5]. In the initial 1–3 years, new memories are impaired; orientations, such as time and place, deteriorate; and personality changes including depressive states to euphoria or excitability occur [6–10]. In the middle 2–10 years, memory impairment progresses to include aphasia, in which the individual cannot understand words and cannot come out; agnosia, in which they cannot understand what is in front of them; and apraxia, which affects wearing clothes and other tasks [11–13]. In the latter 8–12 years, the number of words decreases, the limbs become stiff, and the patient becomes bedridden. Death usually occurs after 10–15 years, from systemic complications, such as pneumonia [14]. AD is characterized neuropathologically by atrophy of the hippocampus and cerebral cortex and microscopically by extensive neuronal loss, senile plaques, and neurofibrillary tangle deposition [15–18]. The major components of senile plaques and neurofibrillary tangles have been identified as an amyloid β protein (Aβ) and a highly phosphorylated

tau protein, respectively [16,19]. The senile plaques, which are Aβ deposits, are more disease-specific for AD than neurofibrillary tangles. Diffused senile plaques, which are predominantly non-fibrotic Aβ deposits, are also the earliest lesions in the AD brain. In addition, point mutations and duplications of the amyloid precursor protein (APP) have been found to be linked to the disease in a familial AD (FAD) with autosomal dominant inheritance [20,21]. Furthermore, Aβ, particularly polymerized Aβ aggregates, are neurotoxic. The Aβ-centered hypothesis about the pathogenesis of AD is called the amyloid cascade hypothesis [22]. Development of treatment methods based on the amyloid cascade hypothesis, such as β-secretase inhibitors, γ-secretase inhibitors, activation of α-secretase and Aβ-degrading enzymes, Aβ immunotherapy, Aβ aggregation inhibitors, anti-inflammatory drugs, and neuroprotective drugs, is in progress [23–29]. In addition, soluble Aβ oligomers, which are polymerized in small amounts, are considered to be the major pathogenesis-related substances rather than the highly polymerized Aβ deposited as senile plaques [30]. Although the cause of AD is not yet fully understood, it is becoming increasingly clear that it is caused by a complex series of events that occur in the brain over a long period of time. The cause is considered to be a combination of multiple factors, such as heredity, environment, and lifestyle [31,32]. This review presents the summary of the potential of oral vaccine therapy for Alzheimer's disease.

## 2. Structural Relationship between Aβ and AD

Aβ is a type of protein produced in the brain of healthy people, wherein it is normally decomposed and excreted in a short period of time. However, when Aβ peptides sticks together to form spots called senile plaques, the protein becomes abnormal and toxic and is not excreted; it then accumulates in the brain, and clings to healthy nerve cells [33]. Thereafter, the toxins produced by Aβ kill nerve cells, making it impossible to transmit information; the brain gradually shrinks, and as a result, Alzheimer's dementia progresses. It has been found that the accumulation of Aβ gets accelerated with the simultaneous presence of cerebral arteriosclerosis [34]. The accumulation of Aβ occurs for over 10 years before the onset of dementia, and there are two main reasons for its accumulation. The first is a lack of exercise. Studies have shown that exercise reduces the accumulation of Aβ [35]. Conversely, people who do not exercise regularly are likely to be in a state that promotes Aβ accumulation. The second is reduced cognitive activity. Neglecting to use cognitive functions, such as thinking, remembering, and judging on a daily basis, can lead to the accumulation of Aβ, which can lead to a decline in the cognitive function. A comparison between rats bred in a highly stimulating environment and those bred in a monotonous, low-stimulating environment showed that the former accumulated less Aβ [36]. In addition, genetically engineered mice with a mutant APP, which was discovered in FAD, were observed to induce amyloid deposits resembling senile plaques, and are attracting attention as an AD animal model [37].

The "Aβ Hypothesis," which states that the toxicity of Aβ accumulated in the brain causes neuronal cell death, brain atrophy, and the onset of dementia has attracted attention as the most influential theory since it was proposed in 2010, and it is also the mainstay in the development of new drugs for AD [38]. Therefore, it may be expected that dementia may be improved by promoting the excretion of the deposited Aβ. Another cause of AD is neurofibrillary tangles. These form when healthy nerve cells become entangled with a substance called tau protein, and similar to Aβ, accumulation of unnecessary substances impairs the function of nerve cells and causes them to die [39,40]. Thus, when Aβ and neurofibrillary tangles cause neurons to die one after another and stop working normally, the brain atrophies, which is the mechanism of dementia. The United States Food and Drug Administration has approved some drugs for the treatment for AD. Donepezil, rivastigmine, and galantamine are used to treat mild to moderate AD [41–44]. Donepezil is also available for advanced AD, and memantine is used to treat moderate-to-severe AD. These drugs work by controlling neurotransmitters, which are chemicals that carry information between neurons and help maintain thinking, memory, and speech, thus

helping improve certain behavioral and psychiatric symptoms. However, these drugs do not change the progression of AD itself; they may or may not work on some people and may only work for a limited period of time. In addition, because these drugs act similarly, switching from one drug to another is not believed to produce significantly different results. Moreover, among the known risks of AD, since lifestyle-related diseases, such as diabetes [45–51] and hypertension [52–57] are listed first, there are also numerous results that these therapeutic agents prevent AD.

In contrast, as an AD treatment based on the amyloid hypothesis, efforts are focused on prevention rather than treatment, and vaccines are attracting attention as a way to prevent the accumulation of Aβ. Certain mutant APP-expressing mice show Aβ deposition in the brain as they age, but repeated subcutaneous injections of a mixture of Aβ and an adjuvant at a young age resulted in the detection of anti-Aβ antibodies in their sera, and Aβ deposition did not recur even in old age [58–66]. Injection of a mixture of Aβ and an adjuvant into a one-week-old mouse, in which Aβ deposits had already started to form in the brain, partially eliminated the formed Aβ deposits. There are two possible mechanisms by which Aβ inoculation reduces Aβ deposition. One possibility is that antibodies produced in the body bind to Aβ and inhibit the formation of aggregates. Suppression of brain Aβ deposition by peripheral administration of anti-Aβ antibodies supports this idea [67]. Another possibility is that cellular immunity against Aβ was activated. Phagocytosis of deposited Aβ by activated immune system cells may lead to the disappearance of amyloid plaques, suggesting that vaccine therapy may be superior to simple antibody administration [68]. The vaccine efficacy was tested using an animal model and a learning test that correlated Aβ deposition with memory impairment. Aβ inoculation suppresses age-related learning deficits and brain Aβ deposition in a reference memory water maze test in mutant APP-expressing mice [69]; using a different strain of mutant mice and the working memory radial water maze test similarly prevented learning deficits. In this review, we summarise Aβ-based vaccine therapies and novel immunotherapies.

## 3. Molecular Mechanisms of Pathogenesis in AD

APP is a cell membrane receptor-like glycoprotein, a type I single-pass transmembrane protein, wherein Aβ is a segment that spans its transmembrane portion and extends into the extracellular domain (Figure 1) [70]. Polymerization experiments with synthetic Aβ peptides have showed that Aβ42 (43), which terminates at the C-terminal position 42 (43), is significantly more prone to aggregation than Aβ40, which terminates at the C-terminal position 40 [71,72]. Furthermore, once Aβ42(43) aggregates, it triggers the polymerization of not only Aβ42(43) but also Aβ40. Because the accumulation of Aβs, especially long C-terminal Aβ42 and Aβ43 (long Aβs) in the brain, is an important step in the pathogenesis of AD, the accumulation of long Aβs is a common pathology in all types of AD. Since long Aβs tend to aggregate to a higher degree than Aβ40, which normally exists predominantly, Aβ40 is expected to form fibrils around them and form the core of the senile plaques. Furthermore, transgenic mice in which Aβ production is promoted exhibit neuronal degeneration and cognitive impairment similar to those in AD, indicating the primary role of Aβ in AD [73,74]. In addition, since aggregated Aβs cause neurotoxicity, it is expected that long Aβs, which are more likely to aggregate, are more toxic [75]. APP is widely expressed in vivo, and Aβ is produced in most cells as a normal metabolite of APP. β-secretase is a proteolytic enzyme that cleaves Aβ at the N-terminal Asp+1, and this cleavage produces two types of fragments: a membrane-bound C-terminal fragment (CTF) consisting of the extracellular domain of secretory APP (sAPP), called sAPPβ; and a 99-amino acid segment called C99 [76,77]. After being cleaved by β-secretase, C99 acts as a substrate for the second secretase, γ-secretase, and as the C-terminus of Aβ is cleaved, Aβ is secreted outside the cell [78,79]. The CTF is translocated to the nucleus as APP intracellular domain and acts as a transcription factor [80]. Intramembrane cleavage of CTF by γ-secretase first occurs near the cytoplasm and then proceeds through N-terminal cleavage by three amino acids, ultimately producing Aβ. Due to the difference in the

first cleavage site, the C-terminus of Aβ is not singular but forms a spectrum containing 37–43 amino acid residues [81]. In normal cells, Aβ40 predominates, while Aβ42 accounts for only approximately 10% of the total Aβ. A third secretase, α-secretase, cleaves APP at Leu+17 in the middle of Aβ and produces an extracellular domain of sAPP, called sAPPα, and a membrane-bound CTF, called C83, consisting of 83 amino acids, which is further cleaved by γ-secretase into a 3 kDa fragment, called p3 [82–84]. Mutations in APP that cause early-onset of FAD with autosomal dominant inheritance are located near the site of action of secretase and directly affect the degradation efficacy and site of action of secretase. For example, the Swedish APP mutation is a double mutation of LysMet just before the N-terminus of Aβ to AsnLeu, which increases the substrate degradation efficacy of β-secretase and increases Aβ production [85]. Several FAD mutations have also been identified near the action site of γ-secretase, which shifts the action site of γ-secretase to increase the production of Aβ42 [86]. FAD mutations also exist near the action site of α-secretase, and they suppress the degradation efficacy of α-secretase, resulting in increased APP as a substrate for β-secretase, acting in the direction of enhancing Aβ production and aggregation [87]. In addition, FAD families with Osaka mutant APP (E693Δ) produce mutant Aβ (E22Δ), but this mutant form of Aβ mainly forms Aβ oligomers instead of Aβ fibrils, and the toxicity of Aβ oligomers alone causes neurological symptoms [88–90]. In addition, Aβ oligomers were detected in the brain of Osaka mutant APP (E693Δ) transgenic mice, wherein all AD pathologies and cognitive impairments were observed, except for senile plaques [88]. In addition, the same mutation impairs the function of intracellular organelles, including mitochondria, and induces apoptosis, strongly supporting the hypothesis that Aβ oligomers are neurotoxic.

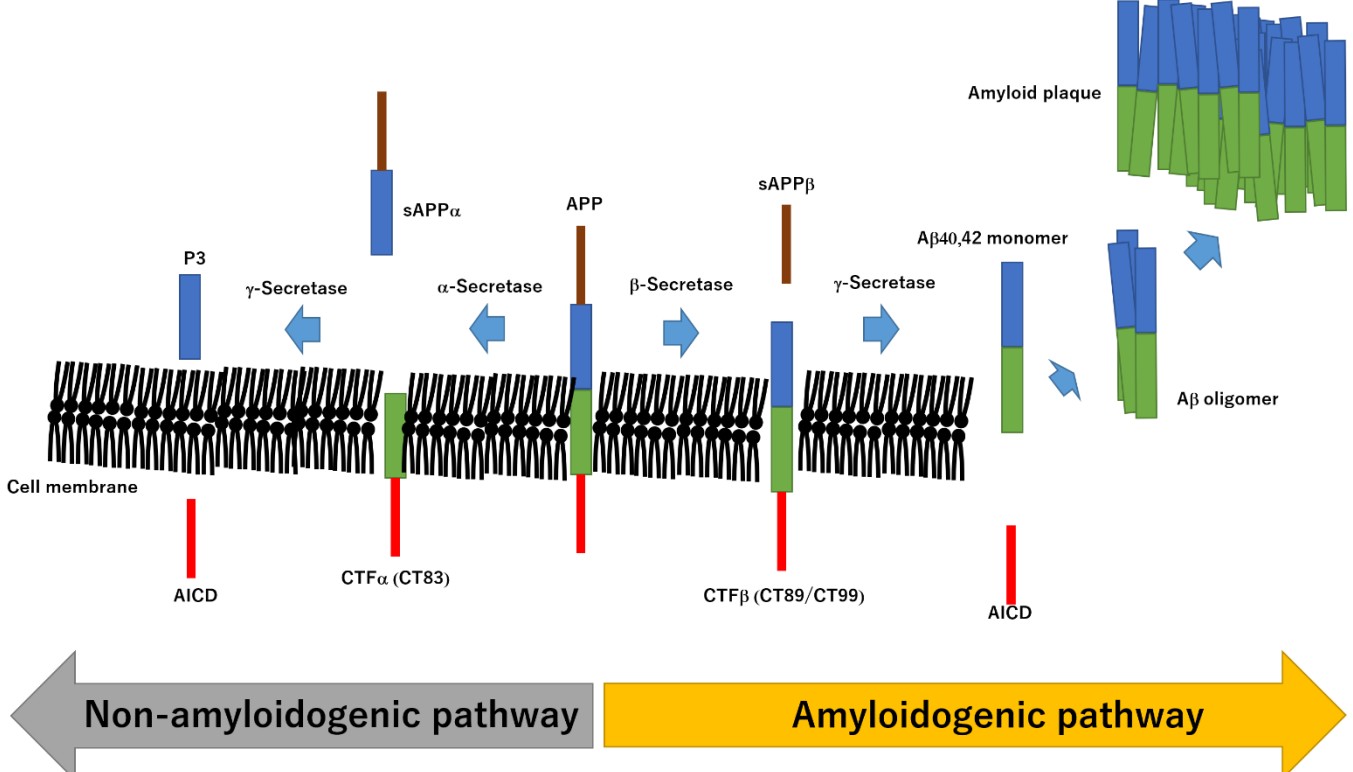

**Figure 1.** Amyloidogenic pathway and non-amyloidogenic pathway. sAPP—soluble amyloid precursor protein, Aβ—amyloid β, AICD—APP intercellular domain, CTF—C-terminal fragment, P3—P3 peptide.

Senile plaques, which are extracellular deposits, are giant insoluble aggregates composed of Aβ fibrils. Addition of Aβ fibrils to cultured neurons causes cell death, the cascade of Aβ fibrils in the amyloid cascade hypothesis was initially considered the reason for

neuronal cell death, causing to cognitive decline [91]. However, the Aβ concentration required for cell death induction by Aβ fibrils is too high, and there is no correlation between AD cognitive impairment and senile plaque density [81]. Therefore, the Aβ fibril amount and difficulties associated with this cascade have been pointed out. Later, the amount of soluble Aβ correlated with the severity of AD, and soluble Aβ aggregates, Aβ oligomers generated at physiological Aβ concentrations, impaired synaptic function, that is the Aβ oligomer hypothesis [92]. Aβ oligomers are classified according to their size into low-n oligomers consisting of 2–3 molecules, Aβ-derived diffusible ligand (ADLL) of approximately 12 molecules, Aβ×56, and protofibrils close to fibrils of 50 molecules or more [93–96]. The produced Aβ molecules immediately form low-n oligomers, such as dimers and trimers, which aggregate low molecular weights. In addition, Aβ dimers have been detected in the soluble fraction of AD and Down syndrome, which is a trisomy of chromosome 21 with APP, brain [97]. ADDLs and Aβ×56 are spherical oligomers with a diameter of 4.8–5.7 nm, molecular weight of 56 kDa, approximately 12 molecules, and a 12-fold increase over healthy controls in the soluble fraction of AD brain.

Aβ protofibrils are fibrillar oligomers with a diameter of 6–10 nm, length of 5–160 nm, and an average molecular weight of 100 kDa or more. It is speculated that Aβ oligomers impair synaptic function from outside the cell via glutamate transporters, insulin receptors, and acetylcholine receptors on the plasma membrane or by inhibiting N-methyl-D-aspartate receptors. In addition, they may induce apoptosis by forming ion channel-like pores in the cell membrane or by accumulating in intracellular organelles and mediating endoplasmic reticulum stress. Moreover, Aβ oligomers impair learning and memory functions by suppressing long-term potentiation (LTP) and enhancing long term depression (LTD), which is closely related to glutamate [98–100].

Oxidative disorders of lipids, proteins, and nucleic acids are observed in the AD brain. Disruption of the balance between oxidative damage, caused by active oxygen and nitrogen, and the antioxidant action that protects the body is expected to be related to the pathology of AD [101]. Juvenile AD has genetic mutations associated with the Aβ production pathway, the main component of senile plaques. These mutations are in the presenilin 1, presenilin 2, and amyloid precursor genes [102–104]. On the contrary, even if there is a mutation in the pathway that phosphorylates tau, which is the main component of neurofibrillary tangles, to fibrillate, results in dementia [105]. Also, apolipoprotein E (APOE) was discovered as a gene responsible for late-onset AD, which develops around the age of 65 [106–110]. There are three isoforms of APOE, and having even one, E4, is likely to cause dementia. E4 homozygotes were found to be 11.6 times more likely to develop AD than E3 homozygotes, implicating genetic predisposition to late-onset AD. In addition, the levels of intracellular Aβs, especially Aβ42, are increased by oxidative stimulation, and the isoform-dependent antioxidant effect of APOE, a risk factor for AD, is involved. Since immunohistochemically, no amyloid fibrils were observed in neurons, this Aβ42 accumulates as soluble monomers of oligomers. Aβ42 accumulated in cells is converted into amyloid β peptide alcohol dehydrogenase (ABAD) and induces free radicals via mitochondrial impairment [111,112]. In addition, intracellular Aβ42 directly binds to the p53 promoter and induces apoptosis by promoting p53 mRNA expression [113]. Based on these findings, Aβ42 accumulated in cells, triggered by overproduction in the endoplasmic reticulum and oxidative stress, causes damage to mitochondria, synapses, and proteasomes, and some translocate to the nucleus to promote p53-dependent apoptosis. In addition, the existence of toxic conformers with turn structure near Glu-22 and Asp-23 instead of non-toxic conformers with turns near Gly-25 and Ser-26 have been shown to be dominant physiologically [114]. This toxic conformer enhances the ability of Met-35 to form radicals, suggesting a relationship with oxidative stress. It was also revealed that MITOL, mitochondria ubiquitin ligase, an enzyme that regulates mitochondrial function, suppresses the production of highly toxic Aβ oligomers [115].

### 4. Alzheimer's Vaccine

Vaccines consist of active immunization, wherein antigens are administered, and passive immunization, wherein antibodies themselves are administered [116]. AD trials of vaccine therapy in humans have raised several concerns. First, in a clinical trial using aggregated Aβ42 as an antigen, the use of purified saponin as an adjuvant caused a strong inflammatory reaction, causing meningoencephalitis in 6% of the 300 participants [117]. It has also been pointed out that there may have been a problem with the surfactant polysorbic acid that enhances the adjuvant effect [118]. A second problem is that most of the trials chose the N-terminus of Aβ as the epitope. The presence of B-cell epitopes for Aβ oligomers at the N-terminus 1–14 of Aβ led to the appearance of an inflammatory Th1 reaction or microhemorrhage [119]. The third problem is that since most of the animal experiments use transgenic mice with marked Aβ accumulation, there are opinions expressing that application on humans should be carried out after using a model in which Aβ slowly accumulates with longevity. In addition, the fact that antibody production against antigens is generally difficult in the elderly is another reason why clinical trials are difficult to succeed. Moreover, it is generally better to administer direct immunization, i.e., DNA vaccination, simultaneously. In contrast, injection immunization and antibody administration have shown that the antibody titer is proportional to the therapeutic effect [120]. However, increasing antibody titer inevitably increases the T cell response. Meningoencephalitis appeared as a side effect of the Alzheimer's vaccine; patients with elevated antibody titers showed little cognitive decline [121]. Therefore, if side effects can be reduced, it forms an effective treatment. Another severe problem is that although it is not possible to predict who will develop dementia, the effect of vaccination cannot be expected, unless it starts before the nerve cells begin to be destroyed. Therefore, a proposal has been made to use amyloid position emission tomography, in which radiolabeled Pittsburgh compound B, a compound that binds to senile plaques, is administered intravenously and then photographed as a biomarker for Aβ accumulation in the brain [122]. In addition, by creating a fusion protein of Aβ with a T-cell epitope, such as hepatitis B surface antigen, tetanus toxin, or diphtheria toxin, which has been administered in the past, fusion proteins, such as Aβ and influenza antigens would be effective antigens, because antibodies would likely be produced against Aβ fragments with B cell epitopes [123].

### 5. Oral Immune Tolerance in Alzheimer's Disease

Absence or suppression of immune response to a specific antigen is called immune tolerance; the immune system does not normally recognize its own antigens, a mechanism known as "self-tolerance" [124]. Immune tolerance exists not only to self-antigens but also to suppress immune responses that are detrimental to the body, such as allergic reactions to food antigens [125]. Oral immune tolerance is a type of immune tolerance that actively suppresses antigen-specific immune responses to antigens encountered in the gastrointestinal tract, such as food antigens, and its breakdown leads to the development of food allergy [126]. Systemic immune responses are observed in mice administered an antigen systemically, such as by subcutaneous injection, followed by another systemic boost with the same antigen. In contrast, when an antigen is first administered orally, the systemic immune response by subsequent systemic boosting is suppressed, that is, oral immune tolerance is induced. To induce intestinal immune tolerance, orally administered antigens must be recognized by the intestinal immune system. The intestinal immune system is broadly divided into gut-associated lymphoid tissue (GALT) and immune-effective tissue [127–129]. GALT has organized structures, such as lymphoid follicles, typified by Peyer's patches and isolated lymphoid tissue [130]. The immune-effective tissues are a group of cells responsible for various immune functions, including the innate immune system cells, such as macrophages, dendritic cells, and innate lymphocytes, which are scattered in the lamina propria, and antibody-producing B cells and effector T cells [129,131,132]. The intestinal immune system contains more than 60–70% of all the peripheral lymphocytes, and the intestinal tract can be the largest peripheral immune system in the human body.

Orally ingested proteins are digested into peptides and amino acids by the digestive enzymes. However, a small number of them can cross the epithelial cell layer without being digested, reach the intestinal immune system, and trigger an immune response by being taken up by antigen-presenting cells [133–136]. Inhibition of protein digestion is associated with increased induction of IgE to food antigens [137]. Orally ingested antigens are taken into the body through various routes depending on their properties, such as size and solubility, leading to immune tolerance and induction of immunity. M cells are a subset of epithelial cells sporadically present in the follicle-associated epithelium covering the lymphoid follicles of GALT, such as Peyer's patches and isolated lymphoid tissue [138]. These are specialized cells for the uptake of large particulate antigens, such as viruses and bacteria, through phagocytosis. Lysosomes are not developed in M cells, and ingested antigens are transferred directly to the GALT and the dendritic cells by transcytosis to induce an intestinal immune response [127,139]. In contrast, food-derived soluble proteins, which are significantly smaller than those of bacteria and viruses, are taken up by epithelial cells of the follicle-associated epithelium surrounding M cells rather than by the M cells themselves. IgA produced by GALT against antigens taken up in this way has a neutralizing effect not only on luminal bacteria but also on food antigens and contribute to immune tolerance through immune exclusion [140].

Soluble proteins are absorbed by absorptive epithelial cells and are transported by two routes: the transcellular route, which is mediated by vesicular transport within the cells; and the paracellular route, wherein they are carried between cells [141,142]. In the transcellular pathway, some antigens are degraded by lysosomes, but some are released to the basement membrane by transcytosis. Complexes of antigenic peptides and MHC class II produced by degradation in lysosomes are released from the basement membrane surface on the exosome membrane and are presented by interaction with dendritic cells. In steady state gut, the paracellular pathway of soluble proteins is suppressed by tight junctions, but in the so-called leaky gut state, wherein the intestinal epithelial barrier is weakened, increased transport of intact food-derived antigens via the paracellular pathway is associated with allergenic activity. In addition, goblet cells, which are mucus-producing cells, also play an important role in transporting soluble protein antigens (Figure 2) [143,144]. Low molecular weight soluble antigens injected into the intestinal lumen are preferentially taken up by goblet cells through a phenomenon called goblet cell-associated antigen passages, which closely interacts only with CD103(+) CX3CR1(−) dendritic cells in the lamina propria, which are thought to be involved in the induction of immune tolerance and delivery of antigens [145–151]. In addition to transport through epithelial cells, CX3CR1high macrophages extend dendrites between epithelial cells without disrupting tight junctions, reach the lumen, and take up bacteria and soluble protein antigens. CX3CR1high macrophages normally do not migrate to the mesenteric lymph node, which is the draining lymph node of the intestine, and pass the antigen and peptide MHC II complex, after digestion of the ingested protein, to CD103(+) dendritic cells; the transferred CD103(+) dendritic cells migrate from the lamina propria to the mesenteric lymph node and present antigens to naïve T cells.

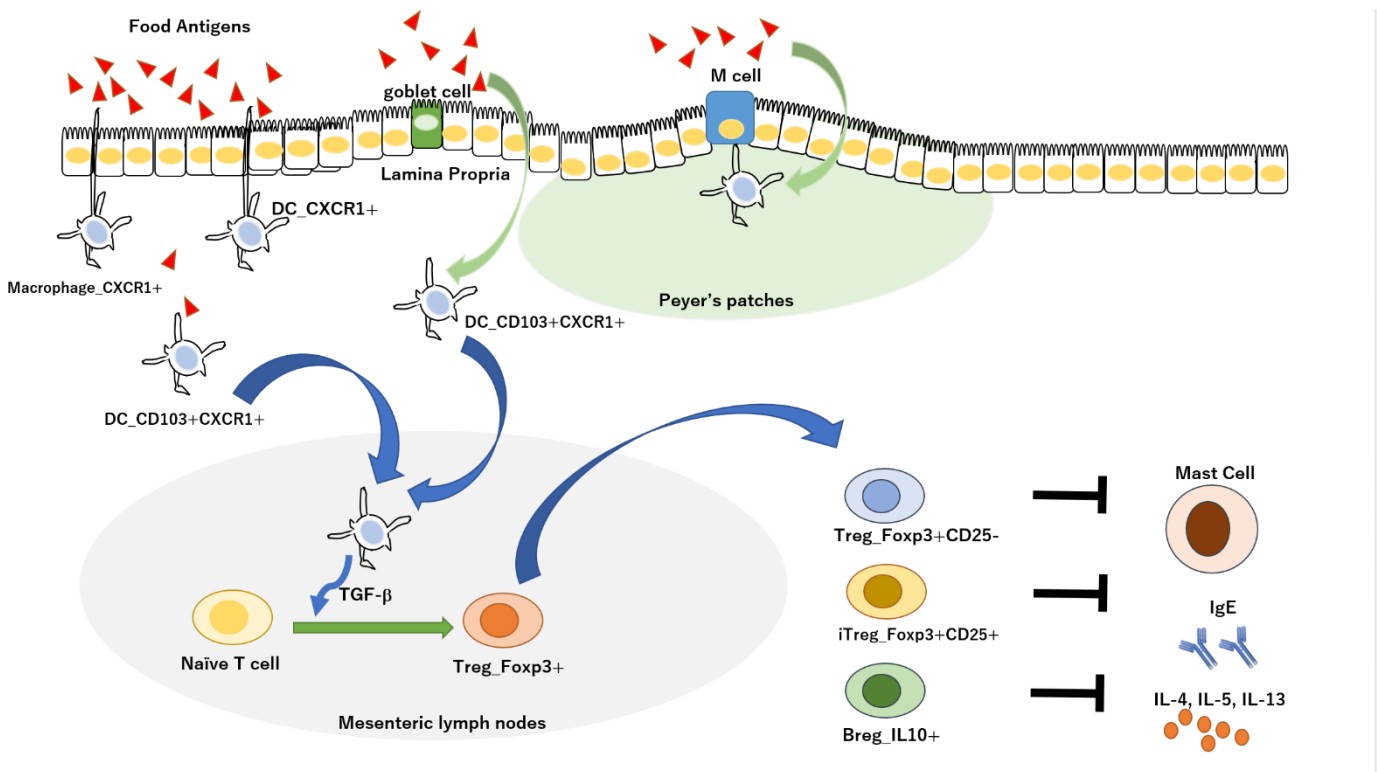

**Figure 2.** Oral tolerance and intestinal immunity to food antigens. In the intestinal immune system, there is a mechanism "oral immune tolerance" that prevents excessive immune response to foreign foods, and those who break through or avoid it cause allergies. Intestinal dendritic cells induce regulatory T (Treg) cells with immunosuppressive potential. In particular, it has a high ability to induce Treg cells that express Foxp3 in mesenteric lymph node dendritic cells (DCs). Among DCs, CD103+DCs have particularly high ability to induce Foxp3+Treg cells. Macrophages receive antigen and pass it on to CD103+DCs.

Oral tolerance can be induced with either a single oral dose of a relatively large dose of antigen or daily oral doses of relatively small doses of antigen. In high-dose tolerance, immune tolerance occurs as a result of antigen-specific T cell anergy and clonal deletion due to orally administered antigens [152]. Anergy induces a "non-response" state in which an immune response does not occur even when stimulated by an antigen. In contrast, clonal deletion causes T cells to undergo cell death by apoptosis by receiving strong antigenic stimulation. Antigens absorbed in the intestinal tract are taken up by plasmacytoid dendritic cells in the mesenteric lymph node or liver, which causes clonal elimination of antigen-specific T cells [153]. Immune responses of antigen-specific T cells that escape clonal deletion are suppressed by peripherally derived regulatory T cells (pTregs) induced in secondary lymphoid tissues. At this time, CD103(+) dendritic cells in the mesenteric lymph node and plasmacytoid dendritic cells in the tonsil, which is the nasopharyngeal lymphoid tissue, act to induce pTregs [154–156]. Macrophages ingesting apoptotic T cells due to high-dose tolerance produce TGF-β and contribute to pTreg induction [157–159]. In contrast, low-dose tolerance induces Tregs [160]. Tolerance is induced by isolating CD4(+) T cells from orally challenged and orally tolerant mice and transferring them to non-immune mice [161]. Conversely, mice treated to deplete CD4(+) T cells do not develop oral tolerance. Taken together, it suggests that among CD4(+) T cells, Tregs are important for establishing oral tolerance. By negatively regulating immune responses and inducing immune tolerance, Tregs are important CD4(+) T cells that suppress the development of diseases caused by abnormal immune responses, such as autoimmune diseases and allergies [162–166]. Furthermore, Foxp3 is a transcription factor essential for

Treg differentiation; Tregs are broadly divided into two subsets based on their mode of differentiation [167]. Thymocytes have with T cell antigen receptors with strong affinity for self-antigens, and immune T cells undergo "negative selection" by stimulation of autoantigens and MHC expressed in thymic medullary epithelial cells and are eliminated by apoptotic cell death [168,169]. However, some autoreactive thymocytes escape negative selection by overexpressing Foxp3 and differentiate into Tregs, which are thymus derived Tregs (tTregs) [170,171]. On the contrary, thymocytes that do not have a strong affinity for autoantigen and MSC mature through positive selection and emerge from the thymus to become naïve T cells. Oral tolerance requires pTregs and not tTregs [172,173]. In addition, CD103+ dendritic cells that have captured antigens in the lamina propria migrate to the mesenteric lymph node and induce pTreg through a mechanism dependent on TGFβ and retinoic acid. Also, mice raised on an antigen-free diet consisting of amino acids show decreased small intestinal pTreg, high serum IgE levels against food antigens, and exacerbation of food allergy symptoms [174,175]. This suggests that food antigens themselves play a role in oral tolerance to food antigens through induction of pTregs in the small intestine. In addition, mice treated with antibiotics under sterile conditions or under reduced gut microbiota diversity exhibit high IgE levels and increased sensitization to food antigens [176,177]. Differentiation of naïve T cells into pTregs in the local colon is mediated by the enhanced expression of the Treg master transcription factor Foxp3 gene in differentiating pTregs due to histone deacetylase inhibitory action of butyric acid, which is a metabolite produced by the bacteria Clostridia in the intestinal flora [178,179]. The induced pTreg suppresses antigen specific IgE in a mouse model of food allergy. In addition, secondary bile acids, which are produced by intestinal bacterial metabolism of primary bile acids secreted by the host into bile, also promote pTreg differentiation in the colon.

Blocking the production of Aβ and preventing its long-term deposition can reduce the risk of Alzheimer's disease. Also, if the deposited Aβ, which slowly deposits in brain tissue over a long period of several decades, leading to neural cell death, can be somehow removed, the population of Alzheimer's disease is expected to decrease dramatically. Administration of anti-Aβ antibodies to Alzheimer's disease model mice can suppress the deposition of Aβ in the brain. However, in a phase II clinical trial with direct administration of Aβ, meningoencephalitis occurred, but subsequent necropsy observed decreased Aβ deposition in the brain. Therefore, considering the possibility of side effects due to direct administration of Aβ, the practical use of the safe oral vaccine, which expressed Aβ in plants, is expected. Indeed, administration of this oral vaccine to Alzheimer's model mice reduced Aβ accumulation in the brain. Moreover, almost no expression of inflammatory IgG was observed. In addition, considering the long-term administration of this vaccine, an oral vaccine using rice was devised. Similarly, in the model mice, in addition to the reduction in senile plaques by this rice Aβ vaccine, an effective improvement of abnormal behavior, which is suppression of spontaneous activity and proportional to the amount of insoluble Aβ in the brain, was also observed. Therefore, the oral vaccines are expected to be safe, effective, and economical vaccines against Alzheimer's disease.

## 6. Conclusions

Since aggregation and accumulation of Aβ in the brain is the cause of AD onset, suppression of Aβ aggregation and efficient removal of aggregated Aβ are considered fundamental treatment strategies for AD, but a radical treatment has not yet been established. For treating AD in the future, Aβ-degrading enzymes, secretase inhibitors, and Aβ vaccine therapy are expected to be effective means. In particular, the Aβ vaccine may be the only means of removing the already deposited senile plaques. Thus, it may be expected that this method will be established as a new fundamental treatment for AD by further improving upon oral immune tolerance and safety of therapeutic application.

**Author Contributions:** Writing—review and editing, Y.M.; supervision, R.Y.; funding acquisition, Y.M. All authors have read and agreed to the published version of the manuscript.

**Funding:** This review was funded by Fukuda Foundation for Medical Technology.

**Conflicts of Interest:** The authors declare no conflict of interest.

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
