# Peer review of "Novel Strategy for Alzheimer’s Disease Treatment through Oral Vaccine Therapy with Amyloid Beta"

_biologics, doi:10.3390/biologics3010003_

Round 1

Reviewer 1 Report

Authors summarized the molecular mechanisms of AD pathogenesis due to Aβ, some AD vaccine studies, and oral immune tolerance. Authors need to discuss the oral vaccine relative to AD (including the literature in this field) and mention the pros and cons of this strategy for AD pathogenesis and progression and compared to AD vaccines administrated by other routes.

Figure 1: Please add a legend simplifying the different steps of the figure and including the abbreviations stated in the figure. Please write “Cell membrane” at the lipid bilayer.

Figure 2: Please add a legend indicating the different steps of the figure.

In the section of “Molecular mechanisms of pathogenesis in AD” I suggest adding a few sentences about the mitochondrial dysfunction caused by APP and Aβ in AD pathogenesis.

Line 345: Please correct “CD103(+)” instead of Cd103(+).

Line 386: Please correct “Aβ” instead of Ab.

Some sentences in the review have 11 references (Ex. Line 112) and 9 references (Ex. Line 128), please reference only the most relevant literature to the information stated in the review.

Author Response

Authors summarized the molecular mechanisms of AD pathogenesis due to Aβ, some AD vaccine studies, and oral immune tolerance. Authors need to discuss the oral vaccine relative to AD (including the literature in this field) and mention the pros and cons of this strategy for AD pathogenesis and progression and compared to AD vaccines administrated by other routes.

Thank you very much for reviewer’s comments. According to the comments, we corrected it in Discission section by adding some sentences on line 397-415, in page 9-10.

Figure 1: Please add a legend simplifying the different steps of the figure and including the abbreviations stated in the figure. Please write “Cell membrane” at the lipid bilayer.

Thank you very much for reviewer’s comments. According to the comments, we corrected fig.1.

Figure 2: Please add a legend indicating the different steps of the figure.

Thank you very much for reviewer’s comments. According to the comments, we corrected fig.2.

In the section of “Molecular mechanisms of pathogenesis in AD” I suggest adding a few sentences about the mitochondrial dysfunction caused by APP and Aβ in AD pathogenesis.

Thank you very much for reviewer’s comments. According to the comments, we corrected it in “3. Molecular mechanisms of pathogenesis in AD” section by adding some sentences and reference on line 239-241, in page 6.

Line 345: Please correct “CD103(+)” instead of Cd103(+).

Thank you very much for reviewer’s comments. According to the comments, we corrected the typo.

Line 386: Please correct “Aβ” instead of Ab.

Thank you very much for reviewer’s comments. According to the comments, we corrected the typo.

Some sentences in the review have 11 references (Ex. Line 112) and 9 references (Ex. Line 128), please reference only the most relevant literature to the information stated in the review.

Thank you very much for reviewer’s comments. According to the comments, we corrected it by reducing number of references.

Submission Date

25 November 2022

Date of this review

09 Dec 2022 17:00:33

Reviewer 2 Report

In this manuscript, Yasunari et al. discussed about the potential, and limitations of oral vaccine therapy for the treatment of Alzheimer’s disease. Overall, the manuscript is quite informative but needs some below mentioned modifications.

1.       Abstract needs to be modified without detailing much on the advantages and disadvantages of the vaccine therapy. Simply mention the potential of the oral vaccine therapy that has been shown by the research and what you are discussing in this review.

2.       Line 62: Authors need to rephrase the statement as the review is not summarizing the mechanisms of AD pathogenesis but rather the potential of oral vaccine therapy.

3.       Line 87-94: This should be merged with section 3.

4.       Line 130-142: Respective lines need proper references.

5.       Section-3: Authors are suggested to include the schematic for the mechanism for proper understanding to readers.

6.       Line 173-176: Need references.

7.       Line 184-187: Rephrase.

8.       Line 202: Expand LTP and LTD

9.       Line 208-210: Rephrase

10.   Line 235-241: This reviewer doesn’t agree with the points mentioned in the context of the advantages of vaccine therapy and the disadvantages of passive immunization. Authors are suggested to review these points and rewrite without proper references.

11.   Line 253: Explain or rephrase ‘direct immunization’.

12.   Line 265-268: Since the fusion protein of Aβ has been made with different antigenic proteins (needs references for each of these), why specifically do authors speculate the fusion protein containing Aβ and influenza antigens is effective? This section merits a neat explanation with proper references.

13.   Section-5: Authors have nicely described the mechanism behind oral immune tolerance however this section doesn’t seem to be linked anywhere with the topic of interest, Alzheimer/ Aβ vaccine therapy. Authors are suggested to add some lines about the oral immune tolerance in case of Aβ oral vaccine therapy.

14.   Line 386: Ab needs to be replaced with Aβ.

Author Response

  1. Abstract needs to be modified without detailing much on the advantages and disadvantages of the vaccine therapy. Simply mention the potential of the oral vaccine therapy that has been shown by the research and what you are discussing in this review.

Thank you very much for reviewer’s comments. According to the comments, we corrected Abstract by adding the potential of the oral vaccine therapy.

  1. Line 62: Authors need to rephrase the statement as the review is not summarizing the mechanisms of AD pathogenesis but rather the potential of oral vaccine therapy.

Thank you very much for reviewer’s comments. According to the comments, we corrected it by correcting as following, the potential of oral vaccine therapy for Alzheimer’s disease.

  1. Line 87-94: This should be merged with section 3.

Thank you very much for reviewer’s comments. According to the comments, we corrected it by moving to section 3.

  1. Line 130-142: Respective lines need proper references.

Thank you very much for reviewer’s comments. According to the comments, we corrected it by adding three references.

  1. Section-3: Authors are suggested to include the schematic for the mechanism for proper understanding to readers.

Thank you very much for reviewer’s comments. According to the comments, we corrected it by moving figure 1 to section 3.

  1. Line 173-176: Need references.

Thank you very much for reviewer’s comments. According to the comments, we corrected it by adding reference [88].

  1. Line 184-187: Rephrase.
  2. Line 202: Expand LTP and LTD

Thank you very much for reviewer’s comments. According to the comments, we corrected it by adding official names.

  1. Line 208-210: Rephrase
  2. Line 235-241: This reviewer doesn’t agree with the points mentioned in the context of the advantages of vaccine therapy and the disadvantages of passive immunization. Authors are suggested to review these points and rewrite without proper references.

Thank you very much for reviewer’s comments. According to the comments, we corrected and rewrote.

  1. Line 253: Explain or rephrase ‘direct immunization’.

Thank you very much for reviewer’s comments. According to the comments, we corrected it.

  1. Line 265-268: Since the fusion protein of Aβ has been made with different antigenic proteins (needs references for each of these), why specifically do authors speculate the fusion protein containing Aβ and influenza antigens is effective? This section merits a neat explanation with proper references.

Thank you very much for reviewer’s comments. According to the comments, we corrected it by adding reference [124].

  1. Section-5: Authors have nicely described the mechanism behind oral immune tolerance however this section doesn’t seem to be linked anywhere with the topic of interest, Alzheimer/ Aβ vaccine therapy. Authors are suggested to add some lines about the oral immune tolerance in case of Aβ oral vaccine therapy.

Thank you very much for reviewer’s comments. According to the comments, we corrected it by adding some sentences as 6. Discussion section.

  1. Line 386: Ab needs to be replaced with Aβ.

Thank you very much for reviewer’s comments. According to the comments, we corrected this typo.

Round 2

Reviewer 1 Report

Thank you for addressing all the comments provided in the first revision of your review.

Author Response

Thank you very much for your review.

Reviewer 2 Report

This reviewer would like to thank the authors for taking care of most of the points.

Along with these, please edit as per the following:

1.       Comment-7: Line 184-187: Rephrase (In edited version line 188-191)

2.       Comment-9: Line 208-210: Rephrase (In edited version 217-220)

3.       This reviewer believes that the discussion section should be merged with section 5 and here the title ‘Oral immune tolerance should be changed to ‘Oral immune tolerance in Alzheimer’s disease'

Author Response

Thank you very much for your review.
